# Differential PaxillinB dynamics at *Dictyostelium* cell-substrate adhesions

Julio C. Fierro Morales and Minna Roh-Johnson*

## ABSTRACT

Adhesion-based migration is regulated by focal adhesions: multi-protein nanostructures linking the intracellular cytoskeleton to the extracellular substrate. Efficient adhesion-based migration has been shown to be regulated by focal adhesion dynamics such as lifetime, size and turnover, which in turn are influenced by the molecular composition of focal adhesions. We recently identified the formation of cell-substrate adhesion populations in *Dictyostelium discoideum* with differing molecular compositions, but it is unclear how these distinct compositions influence *Dictyostelium* adhesion dynamics and cell migration. Here, we further investigate the role of VinculinB – the *Dictyostelium* homologue of Vinculin – localization on *Dictyostelium* adhesion lifetime and protein turnover during cell migration. We show that co-localization of VinculinB to PaxillinB-positive cell-substrate adhesions increases adhesion lifetime without changing PaxillinB turnover. We further show that truncation of the PaxillinB N-terminus, which perturbs VinculinB co-localization to adhesions, surprisingly also increases adhesion lifetime and decreases PaxillinB turnover at adhesions. These findings suggest that similar to mammalian focal adhesions, molecular composition of *Dictyostelium* cell-substrate adhesion regulates their adhesion lifetimes and protein turnover, providing insight into how cell-substrate adhesions function during *Dictyostelium* cell migration.

KEY WORDS: *Dictyostelium discoideum*, Adhesion dynamics, Cell migration

## INTRODUCTION

Cell migration is key for a variety of cellular processes and is often enabled via the use of molecular mechanisms to allow cells to sense and navigate their environment efficiently (Alonso-Matilla et al., 2025; De Pascalis and Etienne-Manneville, 2017; Merino-Casallo et al., 2022; Yamada and Sixt, 2019). One such example is adhesion-based migration, which is enabled via the formation and use of focal adhesions (FAs) – multilayered protein nanostructures that physically link the intracellular actin cytoskeleton to the underlying extracellular matrix or substrate (Case and Waterman, 2015; Gardel et al., 2010; Kanchanawong et al., 2010; Lo, 2006). Cell migration is facilitated by the dynamic nature of FAs and their individual components (Martino et al., 2018; Parsons et al., 2010; Stutchbury et al., 2017). During cell

migration, FAs dynamically assemble and disassemble, and the molecular composition of FAs is influenced by factors such as mechanical stimuli, cytoskeletal reorganization, and protein-protein interactions within the FA (Gupton and Waterman-Storer, 2006; Legerstee et al., 2019; Lele et al., 2008; Oakes et al., 2012; Zamir et al., 1999). The molecular composition of FAs in turn impacts FA dynamics such as size, lifetime, assembly and disassembly, clustering, and orientation (Al-Fahad et al., 2022; Case et al., 2015; Choi et al., 2008; Laukaitis et al., 2001; Legerstee et al., 2019; Nayal et al., 2006; Stutchbury et al., 2017; Webb et al., 2004; Zaidel-Bar et al., 2003, 2007). These FA features both enable and regulate adhesion-based cell migration (Berginski et al., 2011; Kim and Wirtz, 2013; Maheshwari et al., 2000; Meenderink et al., 2010; Rosen and Dallon, 2022), suggesting a tightly-regulated relationship between FA composition and distribution, FA dynamics, and cell migration speed.

While the vast majority of work characterizing the relationship between cell migration and FA dynamics and composition has been done in mammalian models, non-metazoan species have also been shown to form FA-like cell-substrate adhesions with conserved FA molecules for cell migration (Bukharova et al., 2005; Fierro Morales et al., 2025; Parra-Acero et al., 2020; Patel et al., 2008; Pribic et al., 2011; Sebé-Pedrós et al., 2010; Tsujioka et al., 2019). We recently characterized the formation of FA-like cell-substrate adhesions during cell migration in the model Amoebozoan *Dictyostelium discoideum* and found that PaxillinB and VinculinB – the conserved *Dictyostelium* homologues of core FA molecules Paxillin and Vinculin, respectively – localize to cell-substrate adhesions. Interestingly, however, we observed that putative *Dictyostelium* cell-substrate adhesion composition varies, as some PaxillinB-positive cell-substrate adhesions possessed VinculinB, while others lacked VinculinB co-localization. Furthermore, in contrast to findings in mammalian systems (Beningo et al., 2001; Doyle et al., 2015; Gupton and Waterman-Storer, 2006; Kim and Wirtz, 2013; Ntantie et al., 2018), we identified a surprising inverse relationship between the number of PaxillinB-positive/VinculinB-positive cell substrate adhesions and cell migration speed – cells that formed fewer PaxillinB-positive/VinculinB-positive cell-substrate adhesions exhibited increased migration speeds. These results suggest that the molecular composition of *Dictyostelium* cell-substrate adhesions may play a role in regulating cell migration (Fierro Morales et al., 2025). Given the fundamentally different findings for FA number and cell migration speed in mammalian cells versus in *Dictyostelium* cells, and the complex relationship between FA composition, FA dynamics, and cell migration, we sought to dissect the relationship between *Dictyostelium* adhesion composition and dynamics by focusing PaxillinB and VinculinB at cell-substrate adhesions. Quantifying *Dictyostelium* cell-substrate adhesion dynamics such as lifetime and protein turnover will in turn help further elucidate the role that cell-substrate adhesions play during *Dictyostelium* cell migration.

Department of Biochemistry, University of Utah, Salt Lake City, UT 84112, USA.

*Author for correspondence (roh-johnson@biochem.utah.edu)

J.C.F.M., 0000-0003-4788-8636; M.R.-J., 0000-0003-3961-4547

Here, we determined the dynamics of PaxillinB-positive cell-substrate adhesions in the presence and absence of VinculinB. We found that VinculinB co-localization to PaxillinB-positive adhesions increases adhesion duration and lifetime and decreases disassembly rates but does not impact turnover of PaxillinB molecules at *Dictyostelium* adhesions. Surprisingly, this increase in cell-substrate adhesion duration does not correlate with reduced *Dictyostelium* cell migration speed. Additionally, removing the N-terminal domain of PaxillinB – the region required for Paxillin interaction with several FA proteins in mammalian systems (Lopez-Colome et al., 2017) – led to lack of VinculinB co-localization, but increased PaxillinB-positive adhesion lifetime and decreased adhesion assembly and disassembly rates. We also observed decreased PaxillinB turnover. These results suggest that similar to FAs in metazoan systems, *Dictyostelium* cell-substrate adhesion dynamics such as lifetime and protein turnover are influenced by the molecular composition of the adhesion structure. Furthermore, it suggests additional, currently unidentified proteins also regulate adhesion dynamics, potentially in competition with known proteins such as VinculinB. These findings help build an initial understanding of how cell-substrate adhesions are regulated in *Dictyostelium* during cell migration and provide an exciting foundation for future studies investigating the interplay between cell-substrate adhesion composition, adhesion dynamics, and cell migration in *Dictyostelium*.

## RESULTS

We previously found that the number of cell-substrate adhesions positive for both PaxillinB and VinculinB (PaxB+/VinB+) is inversely correlated with migration speed, putting forth a model that PaxB+/VinB+ adhesions serve as a molecular 'brake' for migration (Fierro Morales et al., 2025). This model proposes that PaxB+/VinB+ adhesions exhibit different dynamics than adhesions positive for PaxillinB but lacking VinculinB (PaxB+/VinB−), with the prediction that PaxB+/VinB+ adhesions exhibit increased stability and/or longer lifetime than PaxB+/VinB− adhesions (Fierro Morales et al., 2025). Thus, to investigate and compare the dynamics of PaxB+/VinB+ and PaxB+/VinB− cell-substrate adhesions during *Dictyostelium* cell migration, we utilized super-resolution fluorescent timelapse microscopy to image the two populations of adhesions (Fig. 1A). Using a semi-automated image analysis pipeline we previously utilized for PaxillinB punctae tracking (Fierro Morales et al., 2025), we first tracked the duration – defined as the total number of frames a PaxillinB-positive adhesion was detected – of the two different adhesion populations during cell migration. Consistent with our hypothesis, we found that PaxB+/VinB+ cell-substrate adhesions were significantly longer-lasting than PaxB+/VinB− cell-substrate adhesions (Fig. 1B). We next utilized a previously described method (Stehbens and Wittmann, 2014) to more robustly quantify adhesion lifetime – defined as the time, in seconds, that GFP-Paxillin fluorescence intensity at adhesions is above half of the maximum fluorescence intensity measured during the duration of the adhesion (Stehbens and Wittmann, 2014) – as well as adhesion assembly and disassembly rates. As expected, PaxB+/VinB+ cell-substrate adhesions exhibited a significantly longer lifetime than PaxB+/VinB− cell-substrate adhesions (Fig. 1C). Additionally, while there was no significant difference in assembly rates between the two adhesion populations (Fig. 1D), PaxB+/VinB− adhesions had significantly faster disassembly rates (Fig. 1E), consistent with the decreased lifetimes. Altogether, these results suggest that PaxB+/VinB+ adhesions are longer lived than PaxB+/VinB− adhesions, suggesting that VinculinB may play a role in stabilizing PaxillinB-positive cell-substrate adhesions.

We next asked if there was a correlation between cell-substrate adhesion duration and *Dictyostelium* cell migration speed. Previous work has shown that several FA characteristics influence cell migration speed such as FA number, size, lifetime, turnover rates, clustering, and distribution (Astro et al., 2016; Berginski et al., 2011; Cleghorn et al., 2015; Kim and Wirtz, 2013; Maheshwari et al., 2000; Meenderink et al., 2010; Rosen and Dallon, 2022), and we previously showed an inverse correlation between the number of PaxB+/VinB+ cell-substrate adhesions and *Dictyostelium* migration speed (Fierro Morales et al., 2025). Given the increased duration of PaxB+/VinB+ adhesions (Fig. 1B and C), we hypothesized that we would observe an inverse correlation between cell migration speed and the average duration of cell-substrate adhesions. Thus, we performed correlation analysis of cell migration speed and cell-substrate adhesion duration. We quantified the average duration for all PaxB+ adhesions in individual cells and correlated the average adhesion duration with migration speed. Contrary to our hypothesis, we did not observe a correlation between cell-substrate adhesion duration and cell migration speed (Fig. S1A). To further test if a correlation between cell-substrate adhesion duration and cell migration speed may be specific to a certain population of cell-substrate adhesions, we repeated the correlation analyses using the average duration measurements for either PaxB+/VinB+ adhesions or PaxB+/VinB− adhesions in individual cells. We still observed no significant correlation between cell migration speed and the average duration of either cell-substrate adhesion population (Fig. S1B and C). Given that a given cell also shows substantial variability of individual punctae duration within that cell (Fig. S1D), these data suggest that the duration of cell substrate adhesions alone does dictate *Dictyostelium* migration speed.

## VinculinB does not impact PaxillinB turnover at cell-substrate adhesions

In addition to influencing overall adhesion lifetime, differing adhesion compositions also influence adhesion dynamics by impacting turnover and association/dissociation of molecules at adhesion sites (Berginski et al., 2011; Lele et al., 2008; Stutchbury et al., 2017; Webb et al., 2004). Thus, we next investigated the effect of VinculinB on PaxillinB turnover at adhesions. Vinculin localization to cell-substrate adhesions and interaction with binding partners in mammalian systems have been shown to regulate turnover of binding partners at adhesions (Carisey et al., 2013; Cohen et al., 2006; Humphries et al., 2007). Thus, we used fluorescence recovery after photobleaching (FRAP) (Fig. 2A) to determine PaxillinB recovery halftime ($t_{1/2}$) and mobility [by calculating the immobile fraction ($IM_f$)] at *Dictyostelium* cell-substrate adhesions in the presence and absence of VinculinB. FRAP analysis showed that PaxillinB turnover at cell-substrate adhesions was fast [$t_{1/2}$ value of ~2 s, compared to ~15-45 s in metazoan systems (Legerstee et al., 2019; Ripamonti et al., 2021; Stutchbury et al., 2017; Xue et al., 2023)], and VinculinB co-localization had no significant effect on PaxillinB $t_{1/2}$ value or mobility (Fig. 2B). Altogether, these data suggest that VinculinB co-localization does not impact turnover dynamics of PaxillinB at *Dictyostelium* cell-substrate adhesions.

## Perturbation of the PaxillinB N-terminus ablates VinculinB co-localization, increases adhesion lifetime, and decreases PaxillinB turnover

We previously observed that expression of a truncated PaxillinB molecule lacking the N-terminus half (PaxillinB-LIMsOnly) led to PaxillinB hyper-localization and reduced VinculinB co-localization at cell-substrate adhesions (Fig. 3A) (Fierro Morales et al., 2025). In

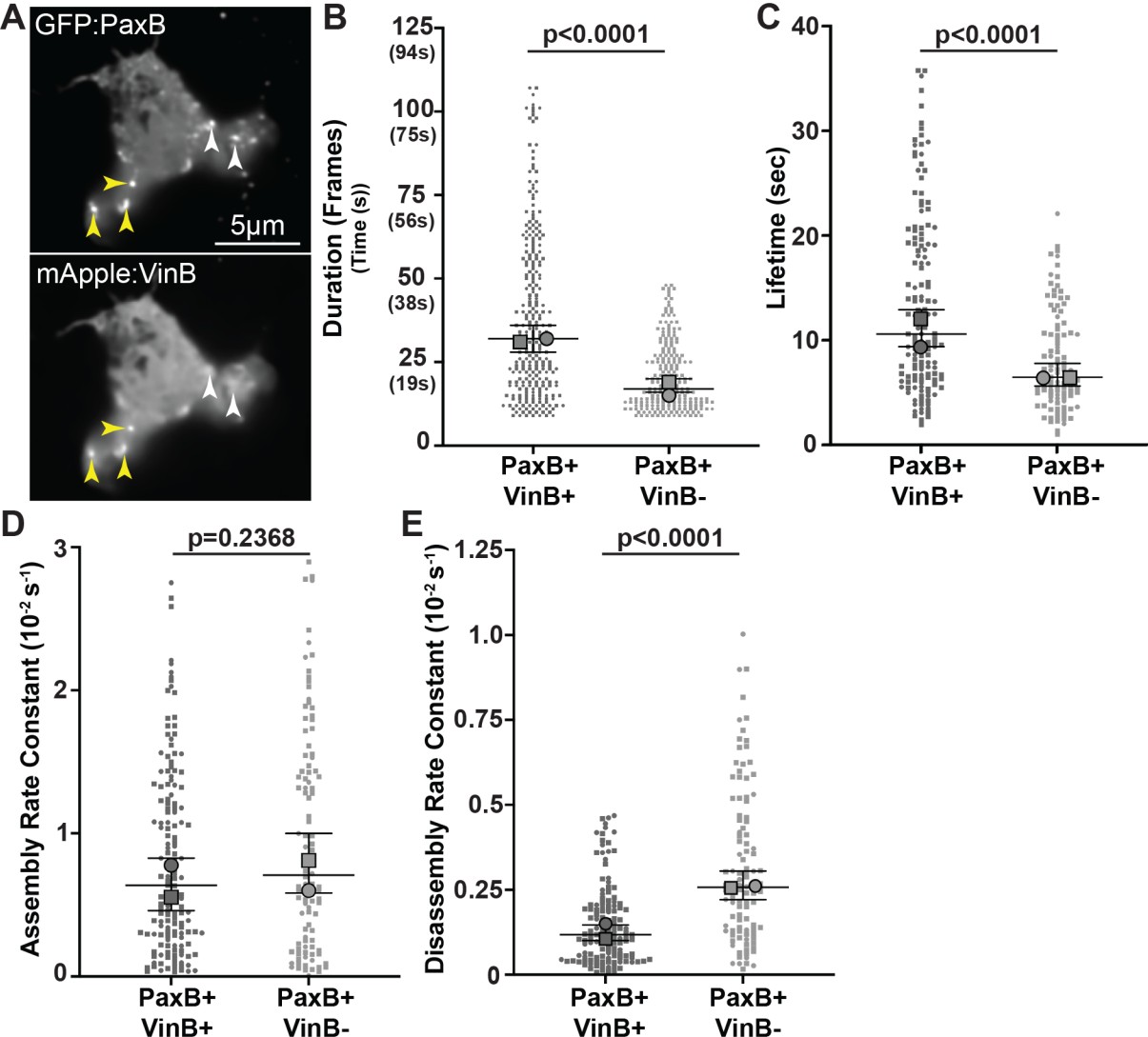

**Fig. 1. Co-localization of VinculinB increases lifetime of PaxillinB-positive *Dictyostelium* cell-substrate adhesions**. (A) Representative fluorescent confocal microscopy images of *paxB⁻/act15*/GFP:PaxB;mApple:VinB *Dictyostelium* cells. Yellow arrowheads indicate cell-substrate adhesions positive for both PaxillinB and VinculinB (PaxB+/VinB+), and white arrowheads indicate cell-substrate adhesions positive for PaxillinB and negative for VinculinB (PaxB+/VinB−). (B) Quantification of PaxB+/VinB+ and PaxB+/VinB− punctae duration (number of frames where each frame was taken every 750 ms). n=266 (PaxB+/VinB+) and 219 (PaxB+/VinB−) punctae across N=2 biological replicates. (C) Quantification of PaxB+/VinB+ and PaxB+/VinB− punctae lifetime. n=145 (PaxB+/VinB+) and 106 (PaxB+/VinB−) punctae across N=2 biological replicates. (D) Quantification of PaxB+/VinB+ and PaxB+/VinB− punctae assembly rate. n=149 (PaxB+/VinB+) and 104 (PaxB+/VinB−) punctae across N=2 biological replicates. (E) Quantification of PaxB+/VinB+ and PaxB+/VinB− punctae disassembly rate. n=135 (PaxB+/VinB+) and 104 (PaxB+/VinB−) punctae across N=2 biological replicates. All data are from punctae tracked during timelapse imaging of *paxB⁻/act15*/GFP:PaxB;mApple:VinB cells. For all graphs, median±95% CI, Wilcoxon rank-sum test.

mammalian models, Paxillin and Vinculin have been shown to directly bind via interaction of the Vinculin tail domain with the Paxillin N-terminal leucine rich aspartate (LD) motifs (Brown et al., 1996; Turner et al., 1990; Wood et al., 1994). Our work and previous work (Bukharova et al., 2005; Fierro Morales et al., 2025) have shown these LD motifs are conserved in *Dictyostelium* PaxillinB. AlphaFold V3 prediction of a binding interaction between VinculinB and PaxillinB predicts that VinculinB binds to PaxillinB at five sites including the conserved LD motif implicated in Paxillin and Vinculin binding in mammalian models (LD2 motif; Fig. 3B). Based on these predicted binding interactions and the lack of VinculinB co-localization with PaxillinB-LIMsOnly, we hypothesized that VinculinB interaction with specifically the PaxillinB N-terminus is necessary for initial binding between the two molecules, with other predicted interactions sites such as the LIM domains functioning to stabilize the interactions.

Given the decreased lifetime of PaxB+/VinB− adhesions compared to PaxB+/VinB+ adhesions (Fig. 1), we hypothesized that PaxillinB-LIMsOnly-positive adhesions may exhibit shorter lifetimes compared to wild-type PaxillinB due to a lack of VinculinB co-localization and binding to PaxillinB. Unexpectedly, however, we observed a statistically significant increase in PaxillinB-LIMsOnly adhesion duration and lifetime compared to wild-type PaxillinB (Fig. 3C and D). Additionally, we observed significantly slower assembly and disassembly rates for PaxillinB-LIMsOnly-positive adhesions compared to those positive for wild-type PaxillinB (Fig. 3E and F). We also used FRAP analysis to investigate PaxillinB turnover and mobility and found that PaxillinB-LIMsOnly molecules at adhesions exhibited a ~2-fold increase in recovery half-times compared to wild-type Paxillin (Fig. 3G, $t_{1/2}$=4.589 compared to $t_{1/2}$=2.398). Furthermore, PaxillinB-LIMsOnly molecules exhibited a

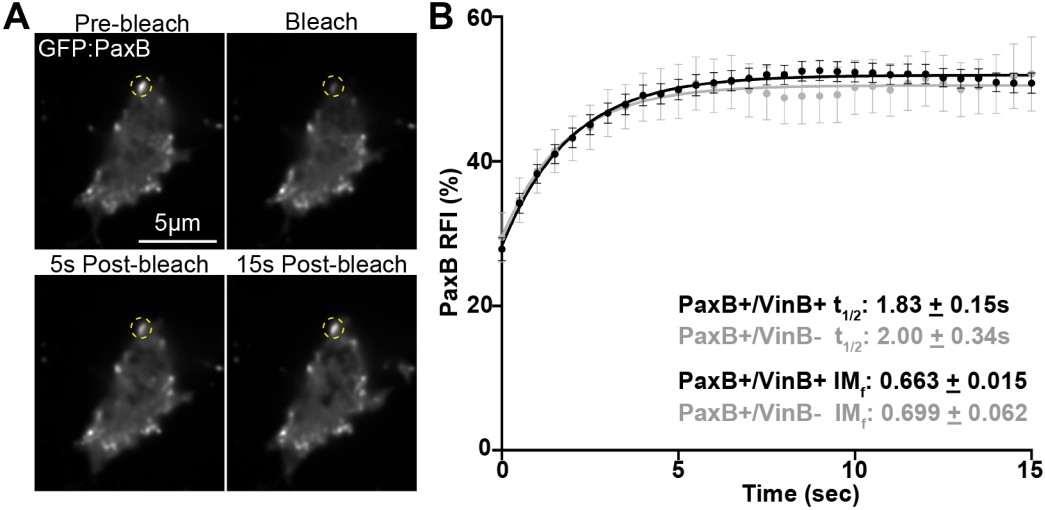

**Fig. 2. PaxillinB turnover at *Dictyostelium* cell-substrate adhesions is not influenced by colocalization of VinculinB.** (A) Representative timelapse fluorescent confocal microscopy images of *paxB⁻/act15*/GFP:PaxB;mApple:VinB *Dictyostelium* cells during FRAP experiments. Yellow dotted circles mark PaxillinB-positive punctae that underwent photobleaching. (B) Cumulative FRAP recovery curves of PaxB+/VinB+ (black) and PaxB+/VinB− (gray) punctae after photobleaching. *n*=69 (PaxB+/VinB+) and 20 (PaxB+/VinB−) punctae across *N*=3 biological replicates. Mean±s.e.m.

larger immobile fraction compared to wild-type PaxillinB (Fig. 3G, $IM_f$=0.746 compared to $IM_f$=0.632). Altogether, these data indicate that despite perturbation of VinculinB co-localization at adhesions, PaxillinB-LIMsOnly-positive adhesions are longer lasting and more stable than wild-type PaxillinB-positive adhesions (Fig. 3G). These findings run in contrast to earlier observations that PaxillinB-positive adhesions lacking VinculinB are shorter lasting than those that possess VinculinB (Fig. 1B and C; Fig. 4). Furthermore, our data show that the PaxillinB N-terminus is important for regulating its turnover at adhesions (Fig. 4).

## DISCUSSION
In this study, we investigated how VinculinB co-localization at PaxillinB-positive cell-substrate adhesions during *Dictyostelium* cell migration impacts the dynamics of PaxillinB-positive adhesions. As expected, we found that PaxillinB-positive cell-substrate adhesions with co-localized VinculinB showed increased adhesion lifetime and duration, and decreased disassembly rates compared to adhesions without VinculinB. However, the presence or absence of VinculinB did not influence the turnover of PaxillinB molecules at adhesions. Interestingly, we also observed that perturbing VinculinB co-localization by expressing a truncated form of PaxillinB (PaxillinB-LIMsOnly) increased adhesion lifetime and duration, while decreasing assembly and disassembly rates. Furthermore, we observed decreased turnover of PaxillinB-LIMsOnly at adhesion sites compared to wild-type PaxillinB. Altogether, these findings highlight that the molecular composition of *Dictyostelium* cell-substrate adhesions influences adhesion dynamics during cell migration.

While these results are a promising step toward furthering our knowledge of how *Dictyostelium* cell-substrate adhesions are regulated, they also posit new questions to consider. One interesting paradox that arises from these studies was that we observed decreased adhesion lifetime when VinculinB did not co-localize at wild-type PaxillinB-positive adhesions (Fig. 1B and C); however, we also observed increased adhesion lifetime when VinculinB was not present at PaxillinB-LIMsOnly-positive adhesions due to perturbation of VinculinB adhesion localization. It is important to note that the presence or absence of VinculinB at adhesions is based on mApple:VinculinB localization, and thus we cannot rule out localization of

endogenous VinculinB when mApple:Vinculin fails to localize at adhesions. Nevertheless, our results suggest that while VinculinB co-localization and putative interaction with PaxillinB plays a role in regulating PaxillinB dynamics at adhesions, it is likely not the sole factor regulating the lifetime of PaxillinB-positive adhesions. Accordingly, while Paxillin and Vinculin are known to interact with one another in mammalian models, Paxillin also serves as a key scaffold for and interacts with several other proteins at focal adhesions through both its N-terminal LD motifs and C-terminal LIM domains to influence adhesion dynamics (Bhattacharya et al., 2025; Bottcher et al., 2017; Brown et al., 1996; Brown and Turner, 2002; Cortesio et al., 2011; Lopez-Colome et al., 2017; Meenderink et al., 2010; Ripamonti et al., 2021; Turner et al., 1999; Xue et al., 2023; Zaidel-Bar et al., 2007). Given the highly conserved nature of these pertinent Paxillin motifs and domains in *Dictyostelium* PaxillinB, we hypothesize that blunt truncation of the entire N-terminal portion of the protein in PaxillinB-LIMsOnly is likely perturbing interactions with other unidentified proteins that regulate PaxillinB localization to and turnover at cell-substrate adhesions during *Dictyostelium* migration, leading to increased PaxillinB lifetime and decreased turnover at adhesions (Fig. 4). Indeed, previous work in a mammalian system has shown that both the N-terminus and C-terminus of Paxillin regulate Paxillin stability and dynamics at FAs (Ripamonti et al., 2021).

We propose possible molecular explanations for these observations. One hypothesis is that there are stability factors recruited to cell-substrate adhesions upon conformation changes in PaxillinB, thereby increasing PaxillinB lifetime at adhesions. These PaxillinB conformational changes could be induced by VinculinB binding as well as the N-terminal truncation in the Paxillin-LIMsOnly mutant. Consistent with this hypothesis, in metazoan systems, Vinculin adopts and induces conformational changes at FAs upon interacting with binding partners such as Talin and Paxillin (Chorev et al., 2018; Cohen et al., 2006; Deakin and Turner, 2011). Conversely, another possibility is the recruitment of disassembly factors that negatively regulate PaxillinB lifetime at cell-substrate adhesions and promote PaxillinB turnover through interactions with the PaxillinB N-terminus. Under this model, the disassembly factors compete with VinculinB binding at the

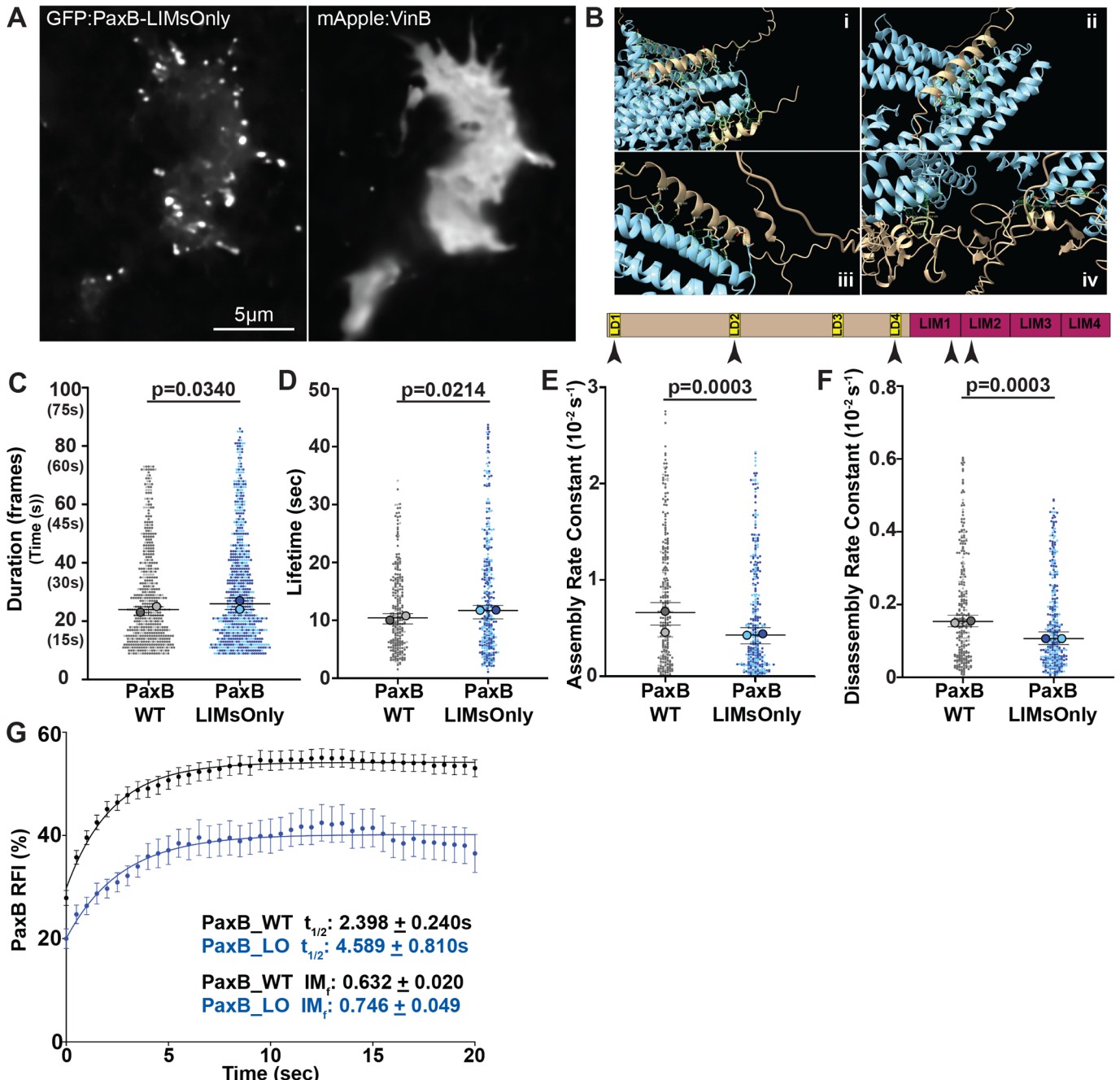

**Fig. 3. PaxillinB _N_-terminus truncation increases adhesion lifetime and decreases PaxillinB turnover at adhesions.** (A) Representative fluorescent confocal microscopy images of _paxB⁻/act15_/GFP:PaxB-LIMsOnly;mApple:VinB _Dictyostelium_ cells. (B) Visualization of putative PaxillinB (in tan) and VinculinB (in light blue) binding interactions as predicted by AlphafoldV3. VinculinB is predicted to interact with the PaxillinB LD1 Motif (i, lower tan helix), LD2 motif (ii), LD4 motif (iii) and residues in the PaxillinB LIM1 and LIM2 domains (iv). Simplified PaxillinB protein domain organization is at the bottom, with black arrowheads indicating predicted sites of VinculinB interaction. Based on the predicted interactions sites and lack of VinculinB co-localization with the PaxillinB-LIMsOnly molecule, we hypothesize the predicted interaction sites at the N-terminus are necessary for VinculinB to bind to PaxillinB, while the predicted sites at the LIM domains alone are not sufficient for binding. (C) Quantification of PaxB-WT and PaxB-LIMsOnly punctae duration. _n_=511 (PaxB-WT) and 659 (PaxB-LIMsOnly) punctae across _N_=2 biological replicates. (D) Quantification of PaxB-WT and PaxB-LIMsOnly punctae lifetime. _n_=267 (PaxB-WT) and 332 (PaxB-LIMsOnly) punctae across _N_=2 biological replicates. (E) Quantification of PaxB-WT and PaxB-LIMsOnly punctae assembly rates. _n_=274 (PaxB-WT) and 328 (PaxB-LIMsOnly) punctae across _N_=2 biological replicates. (F) Quantification of PaxB-WT and PaxB-LIMsOnly punctae disassembly rates during timelapse imaging of _paxB⁻/act15_/GFP:PaxB and _paxB⁻/act15_/GFP:PaxB-LIMsOnly cells, respectively. _n_=273 (PaxB-WT) and 324 (PaxB-LIMsOnly) punctae across _N_=2 biological replicates. For (C-F) All data are from punctae tracked during timelapse imaging of _paxB⁻/act15_/GFP:PaxB and _paxB⁻/act15_/GFP:PaxB-LIMsOnly cells; median±95% CI, Wilcoxon rank-sum test. (G) Cumulative FRAP recovery curves of PaxB-WT (black) and PaxB-LIMsOnly (blue) punctae after photobleaching. _n_=69 (PaxB-WT) and 57 (PaxB-LIMsOnly) punctae across _N_=3 biological replicates, mean±s.e.m.

N-terminus of PaxillinB. In the absence of VinculinB, the disassembly factors bind to PaxillinB, reducing PaxillinB lifetime at adhesions compared to when VinculinB is present. In the PaxillinB-LIMsOnly mutant, both the disassembly factors and VinculinB cannot bind, increasing PaxillinB lifetime compared to wild-type PaxillinB. One potential protein of interest that could

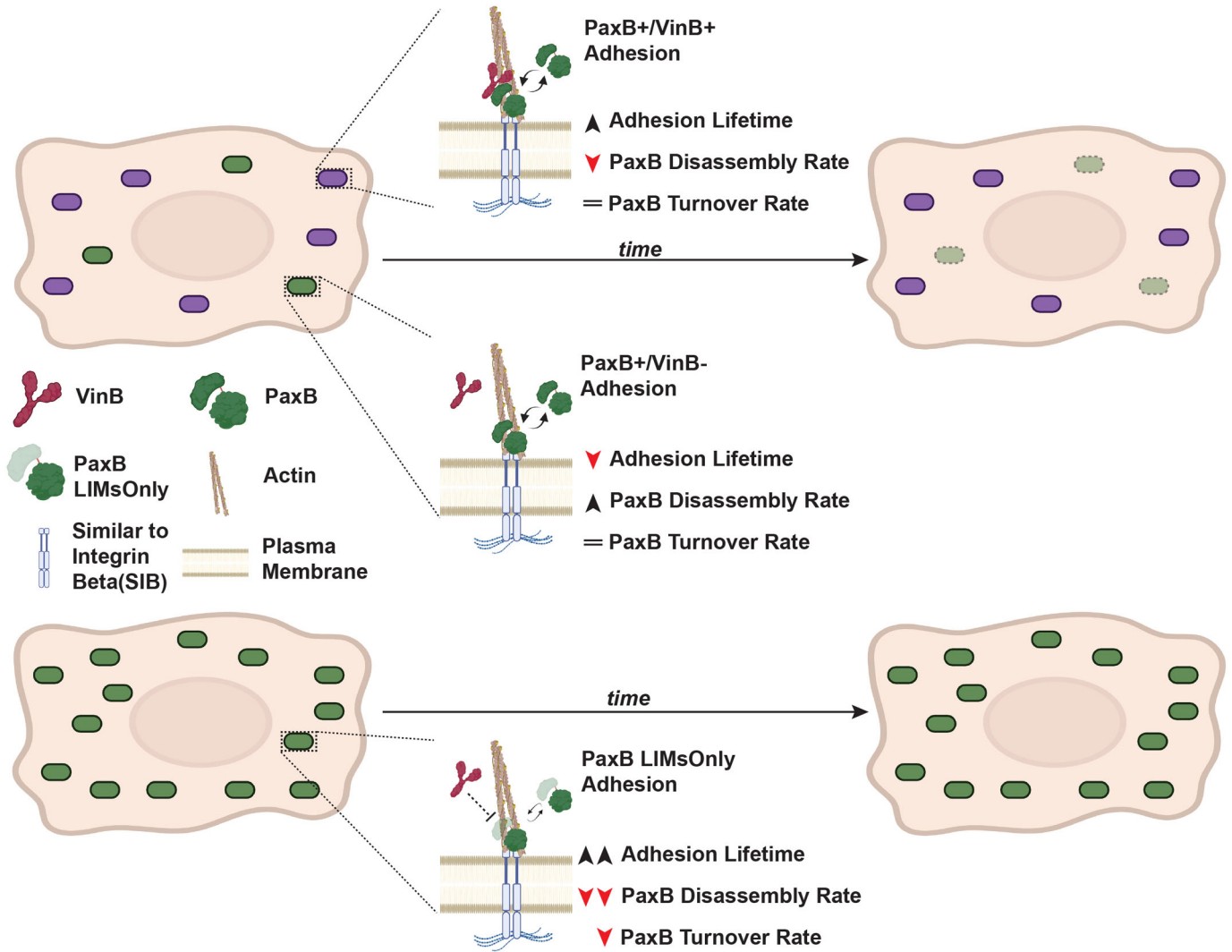

**Fig. 4. Summary of PaxillinB dynamics in the presence/absence of VinculinB and in the PaxillinB-LIMsOnly mutant.** (Top row) When VinculinB co-localizes to PaxillinB-positive adhesions, we observe increased PaxillinB lifetime at adhesions and decreased adhesion disassembly, but no change to PaxillinB turnover. (Middle row) When VinculinB does not co-localize to PaxillinB positive adhesions, PaxillinB lifetime at adhesions is decreased through increased disassembly rates, and we do not observe changes to PaxillinB turnover. (Bottom row) When PaxillinB lacks its N-terminus (PaxillinB-LIMsOnly), VinculinB does not localize to adhesions, and PaxillinB lifetime increases. We further observe decreased PaxillinB turnover with slower recovery times and a larger immobile fraction.

serve as a disassembly factor is FrmA, which has previously been implicated in negatively regulating PaxillinB-positive cell-substrate adhesion turnover and number (Patel et al., 2008), though its exact mechanism of action is currently unclear.

These two models are not mutually exclusive, and it is likely that many proteins regulate the stability of PaxillinB at cell-substrate adhesions through multiple mechanisms. These mechanisms allow *Dictyostelium* to efficiently regulate adhesion dynamics during cell migration, particularly given how dynamic *Dictyostelium* cell-substrate adhesions are in comparison to mammalian focal adhesions (Abramson et al., 2024; Fierro Morales et al., 2025; Han et al., 2021, 2015). Further investigation into identifying more *Dictyostelium* cell-substrate adhesion proteins, as well as characterizing different molecular compositions of adhesions, will help elucidate how *Dictyostelium* cells dynamically form and regulate adhesions to enable efficient cell migration.

Additionally, while our work sheds light on how *Dictyostelium* cell-substrate adhesions are regulated during cell migration, there are some limitations that need to be considered. While the observed

differences in adhesion lifetime and duration between wild-type PaxillinB and PaxillinB LIMsOnly-positive adhesions are statistically significant, the differences are modest (~10% difference). This is in contrast to larger changes in focal adhesion dynamics upon adhesion molecule perturbation, including truncation of the Paxillin N-terminus, that have previously been reported in mammalian literature (Han et al., 2021; Lawson et al., 2012; Ripamonti et al., 2021). We provide two points of consideration on these differences. First, while perturbing PaxillinB domains influences adhesion dynamics, it is likely not the sole variable regulating adhesion dynamics. In mammalian models, factors such as adhesion orientation and location, mechanical forces, substrate stiffness, and actomyosin contractility have all been implicated in regulating adhesion dynamics (Alexandrova et al., 2008; Grashoff et al., 2010; Legerstee et al., 2019; Zhou et al., 2017). Work in *Dictyostelium* has also demonstrated that actomyosin contractility and substrate composition influence *Dictyostelium* cell adhesion and migration (Jay et al., 1995; McCann et al., 2014; Tsujioka et al., 2012). A second, more speculative, consideration is that *Dictyostelium* cell adhesion dynamics are much faster (lifetime is

~15-20 s; Fierro Morales et al., 2025) than what has been reported in metazoan literature (lifetime in the scale of minutes; Ripamonti et al., 2021; Xue et al., 2023). Therefore, even modest changes in adhesion lifetime could have biological consequences on *Dictyostelium* cellular behavior, although additional work is needed to test this hypothesis.

Altogether, our work provides novel insight into the role that VinculinB and PaxillinB play in regulating *Dictyostelium* cell-substrate adhesion dynamics during cell migration. Future research identifying more *Dictyostelium* adhesion proteins, their interactions at adhesions, and their effect on adhesion dynamics will help build a more holistic understanding of how molecular composition as a whole influences cell-substrate adhesion formation and regulation during *Dictyostelium* cell migration. This in turn will serve as broader insight for dissecting how the relationship between cell-substrate adhesion composition, adhesion dynamics, and cell migration has evolved across organisms.

## MATERIALS AND METHODS

### *Dictyostelium* cell culturing
*paxB⁻ Dictyostelium* cells (DSC Strain DBS0236728) were obtained from the *Dictyostelium* StockCenter (DSC) and transformed with wildtype GFP: PaxillinB (UniProt KB accession number Q8MML5), GFP:PaxillinB-LIMsOnly and/or mApple:VinculinB (UniProt KB accession number Q54TU2) plasmids as previously described (Fierro Morales et al., 2025; Gaudet et al., 2007). Cells were cultured axenically at 21°C in HL5 media (Fey et al., 2007) supplemented with 300 μg/ml streptomycin sulfate (Gold Biotechnology), 10 μg/ml Blasticidin S (Thermo Fisher Scientific), and 20 μg/mLG418 Sulfate (Thermo Fisher Scientific) on 10 cm plates. Cells expressing mApple:VinculinB were further supplemented with 50 μg/ml hygromycin B (Thermo Fisher Scientific).

### Generation of expression plasmids
Generation of the wild-type GFP:PaxillinB plasmid has been previously described in Nagasaki et al. (2009) and we thank Drs. Masatsune Tsujioka and Taro Q. P. Uyeda (AIST, Tsukuba, Japan) for sharing the GFP:PaxillinB plasmid. Generation of the GFP:PaxillinB-LIMsOnly and mApple: VinculinB plasmids has been previously described in Fierro Morales et al. (2025). All plasmids are available upon request.

### Preparation of *Dictyostelium* cells for imaging
*Dictyostelium* cells were prepared for imaging as previously described (Fierro Morales et al., 2025). Briefly, cells were cultured in HL5 media on 10 cm cell culture plates to a confluency of 1-2e6 cells/ml and a minimum of 1e7 cells were harvested. Cells were centrifuged at 600 rpm for 5 min at 4°C and resuspended in development buffer (DB; 5 mM Na2HPO4, 5 mM KH2PO4, 1 mM CaCl2, 2 mM MgCl2, pH 6.5) to a concentration of 2.5e6 cells/ml. 1e7 total cells were plated in a 6 cm plate and allowed to adhere for 30 min before being rinsed twice with DB and developed for 3.5 h in 2 ml DB. After development, cells were resuspended and 2.25e5 cells in 1 ml of DB were plated and allowed to adhere in a 35 mm glass bottom dish (FD35-100, World Precision Instruments) for 30 min before imaging.

### *Dictyostelium* imaging
Super resolution spinning disc confocal fluorescence microscopy was done using a Nikon PLAN APO λD 60X/1.42 oil immersion objective with1X zoom on a Nikon Ti2 microscope with a Yokogawa CSU-W1 spinning disc confocal scanner unit, Gataca Systems Live-SR super resolution module and Kinetix 22 Scientific CMOS (sCMOS) camera for super resolution spinning disc confocal microscopy with NIS AR software (RRID:SCR_014329). For all microscopy, *Dictyostelium* cells were in DB.

### Adhesion dynamics acquisition and analysis
To obtain PaxillinB-positive adhesion durations, lifetimes, assembly rates and disassembly rates, randomly migrating *Dictyostelium* cells expressing GFP: PaxillinB and mApple:VinculinB concurrently (for PaxillinB+/VinculinB+

versus PaxillinB+/VinculinB− punctae comparisons) or either GFP:PaxillinB or GFP:PaxillinB-LIMsOnly (for wild-type PaxillinB versus PaxillinB-LIMsOnly punctae comparisons) were imaged every 750 ms for 3 min. After image acquisition, NIS-Elements AR software was used to apply a 'Regional Maxima' kernel (Sysko and Davis, 2010) to timelapse videos of individual *Dictyostelium* cells to remove background noise and enhance thresholding. PaxillinB punctae were identified as binary objects using auto-thresholding along with edge smoothing and clean-up to remove noise from signal pixels. Advanced 2D Tracking was then used to track PaxillinB binaries and the mean fluorescent intensity (MFI) across each frame. PaxillinB punctae that were present in the first or last frame video or that were eight frames or shorter were removed from the dataset. Lastly, manual curation was performed to remove any binaries that were noise or clear false positives based on comparison to the original timelapse video.

To quantify PaxillinB punctae duration, the total number of frames a PaxillinB punctae was tracked was counted on Microsoft Excel (v16.43) (RRID:SCR_016137). PaxillinB punctae lifetime, assembly rate and disassembly rate were calculated as previously described in Stehbens and Wittmann (2014). Briefly, generated PaxillinB binaries MFI data was plotted on Excel and for each binary the MFI value from the first frame was subtracted from all other frames. A three-frame running average MFI was calculated to reduce noise and improve curve fitting for lifetime curve generation using the 'Solver' add-in to generate adhesion assembly and disassembly curves based on fit to logistic function and exponential decay, respectively. The steepness of these curves was calculated to obtain the assembly and disassembly rates, respectively. A quantitative lifetime was then calculated based on the t-half of both the assembly and disassembly fit.

### *Dictyostelium* cell migration speed analysis for correlations
Timelapse videos of individual *Dictyostelium* cells migrating every 750 ms for 3 min used for PaxillinB punctae tracking were used for cell migration speed analysis. Migration was tracked using auto thresholding and binary tracking of whole cells in NIS-Elements AR Advanced 2D Tracking software. Final speeds were calculated by averaging the speeds between each time point of the time-lapse video.

### FRAP experiment and analysis
FRAP was performed on a Nikon Ti2 microscope with a Yokogawa CSU-W1 spinning disc confocal scanner unit equipped with a Ti2-LAPP FRAP module. Single Z-plane images of *Dictyostelium* cells expressing GFP: PaxillinB or GFP:PaxillinB-LIMsOnly were taken every 500 ms for FRAP experiments. A region of interest (ROI) of ~1 μm×1 μm was drawn around individual PaxillinB punctae for photobleaching. Three frames were taken pre-bleaching, and 45 s of imaging were acquired post-bleach. Photobleaching was performed using the 488 nm laser at 50% power. For FRAP experiments comparing PaxillinB+/VinculinB+ versus PaxillinB+/VinculinB− punctae, dual-channel images of cells of interest were taken before FRAP acquisition to confirm the presence or absence of VinculinB at bleached PaxillinB punctae prior to photobleaching.

FRAP analysis was performed as previously described in Xue et al. (2023). Briefly, ImageJ software (RRID:SCR_003070) was used to draw a tight ROI around the bleached, individual PaxillinB punctae as well as an unbleached, non-cell area to subtract background as a control. The total fluorescent intensity (integrated density) was measured for both the bleached and background ROIs using ImageJ. Relative fluorescence intensity (RFI) across time frame was then calculated by subtracting the background and normalizing to the first pre-bleach frame using $RFI=(I_t−I_{bgt})/(I_{pre}−I_{bgpre})$, where $I_t$ is the integrated density of the bleached PaxillinB punctae ROI at time point t and $I_{bgt}$ is the integrated density of background ROI at time point t. $I_{pre}$ is the integrated density of the PaxillinB punctae ROI at the initial prebleach timepoint and $I_{bgpre}$ is the integrated density of the background ROI at the initial prebleach timepoint.

To calculate recovery halftimes ($t_{1/2}$) and filter for outliers, the first 15 s (for PaxillinB+/VinculinB+ versus PaxillinB+/VinculinB− punctae comparisons) or 20 s (for wild-type PaxillinB versus PaxillinB-LIMsOnly punctae comparisons) post-bleaching were used for analysis due to issues with consistent loss of signal and/or full adhesion disassembly past these

time points. For each individual PaxillinB punctae, RFI over time was plotted and GraphPad Prism (v10) (RRID:SCR_002798) was used to fit an exponential plateau curve and obtain the rate constant, k, for each curve. Recovery halftimes ($t_{1/2}$) were calculated using $\ln(2)/k$ and any values greater than the total acquisition time (45 s) were removed along with identified outliers (see Graphical representation and statistical analysis). Individual punctae data sets that were removed using this method were subsequently removed from the mean data sets graphed in Figs 2C and 3F.

For all remaining datasets after recovery halftime ($t_{1/2}$) calculation, the $IM_f$ was calculated using $IM_f=1-((FRFI_f-FRFI_0)/(1-FRFI_0))$, where $FRFI_0$ is the calculated RFI value of the fitted curve at t=0 and $FRFI_f$ is the calculated RFI value of the fitted curve at the final time point (15 s for PaxillinB+/VinculinB+ versus PaxillinB+/VinculinB− punctae comparisons or 20 s for wild-type PaxillinB versus PaxillinB-LIMsOnly punctae comparisons).

## Alphafold prediction of PaxillinB and VinculinB binding interaction

PDB files for the predicted Alphafold structures of PaxillinB (AlphaFoldDB accession number AF-Q8MML5-F1-v4) and VinculinB molecules (AlphaFoldDB accession number AF- Q54TU2-F1-v4) were obtained from the AlphaFold Protein Structure Database (Varadi et al., 2022) and a binding interaction was predicted using the AlphaFold V3 server (Abramson et al., 2024). The predicting PaxillinB-VinculinB binding JSON files were visualized in USCF ChimeraX software developed by the Resource for Biocomputing, Visualization, and Informatics at the University of California, San Francisco, with support from National Institutes of Health R01-GM129325 and the Office of Cyber Infrastructure and Computational Biology, National Institute of Allergy and Infectious Diseases (Meng et al., 2023) and the 'Contacts' function was utilized to identify putative interaction sites between PaxillinB and VinculinB.

## Graphical representation and statistical analysis

All graphs were generated from GraphPad Prism (v10). Statistical and normality analyses were performed using Prism (v10, GraphPad). D'Agostino-Pearson, Shapiro-Wilk, Anderson-Darling and Kolmogorov–Smirnov normality tests were run prior to statistical analyses to test for normality of datasets. Outliers were identified and removed using Prism's 'Identify Outliers' tool using the ROUT method with a coefficient Q value of 1% (Motulsky and Brown, 2006) unless indicated otherwise in the Materials and Methods section for specific analyses. N-values of biological replicates, the number of cells, and specific statistical tests used are found in the figure legends. All statistical comparisons are shown on graphs. Correlation analyses were performed by running the simple linear regression analysis function in Prism. All data except the FRAP curves are presented as SuperPlots (Lord et al., 2020). Briefly, each biological replicate (N) is represented by the larger, outlined symbols and is the average of all data points (n) of the same color. For FRAP curves, the cumulative FRAP data points shown in graphs are the mean of all individual FRAP experiments±s.e.m. and the curve is fitted onto the cumulative FRAP data set.

## Acknowledgements

We would like to thank all members of the Roh-Johnson lab for discussions and workshopping during manuscript and model figure preparation; Dictybase (RRID: SCR_006643), an NIH-funded Stock Center for providing *Dictyostelium* strains used in this work and for maintaining this valuable community resource; Margaret Titus and the *Dictyostelium* Stock Center for *Dictyostelium* reagents; and the Cell Imaging Core at the University of Utah for use of Nikon Yokogawa CSU-W1 spinning disc confocal microscope and NIS-Elements AR Software as well as Michael Bridge, Anton Classen, and Xiang Wang for their assistance.

## Competing interests

The authors declare no competing or financial interests.

## Author contributions

Conceptualization: J.C.F.M., M.R.-J.; Data curation: J.C.F.M.; Formal analysis: J.C.F.M.; Funding acquisition: M.R.-J.; Investigation: J.C.F.M.; Methodology: J.C.F.M.; Project administration: M.R.-J.; Supervision: M.R.-J.; Writing – original draft: J.C.F.M.; Writing – review & editing: J.C.F.M., M.R.-J.

## Funding

The work was funded by National Institutes of Health grant R00CA190836 and R37CA247994 (to M.R.-J.). Open Access funding provided by University of Utah. Deposited in PMC for immediate release.

## Data and resource availability

All relevant data and details of resources can be found within the article and its supplementary information.

## First Person

This article has an associated First Person interview with the first author of the paper.

## Peer review history

The peer review history is available online at https://journals.biologists.com/bio/lookup/doi/10.1242/bio.062197.reviewer-comments.pdf

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
