## [Peer Review File · Biology Open]

Differential PaxillinB dynamics at Dictyostelium cell-substrate adhesions

Julio Cesar Fierro Morales and Minna Roh-Johnson
DOI: 10.1242/bio.062197

Editor: Catherine L. Jackson

Review timeline

Original submission:	4 August 2025
Editorial decision:	12 August 2025
First revision received:	4 September 2025
Accepted:	5 September 2025

Original submission

First decision letter

MS ID#: bio.062197

MS Title: Differential PaxillinB dynamics at Dictyostelium cell-substrate adhesions

Authors: JULIO Cesar FIERRO MORALES; Minna Roh-Johnson

Dear Dr Roh-Johnson,

I have now reached a decision on the above manuscript.

The reviewer reports are shown at the bottom of this email or can be accessed, together with a copy of this decision letter, by going to:

*****.

Reviewer 1

Comments for the author

In this work, Fierro-Morales and Roh-Johnson use Dictyostelium as a model organism to study adhesion-based migration. Specifically, they build on their very recently published article that described cell-substrate adhesion composed of the orthologues of Paxillin and Vinculin to dissect the dynamics of these structures, particularly that of Paxillin with regards to the presence or absence of Vinculin.

Using SR-microscopy, they present solid evidence that PaxB⁺/VinB⁺ adhesions last longer than PaxB⁺/VinB⁻ ones. Interestingly, PaxB⁺/VinB⁻ adhesions disassemble faster than PaxB⁺/VinB⁺ adhesions. This suggests a role for VinculinB in stabilizing PaxillinB adhesions. FRAP analysis showed that VinculinB does not influence PaxillinB turnover at adhesions. Using a PaxillinB mutant (N-terminal truncation), the authors argue that PaxB LIMsOnly adhesions display a longer duration and lifetime than with WT PaxB. Finally they observe a strong difference in PaxB LIMsOnly turnover and mobile fraction compared to WT. Therefore, the authors conclude that the PaxB LIMsOnly mutant must have other consequences than just not interacting with VinculinB. They finally propose alternative models of cell-substrate adhesions' dynamics to account for their findings.

Overall, the work is interesting and well conducted. This opens a lot of questions regarding the dynamics of cell-substrate adhesions and will be interesting for the field, particularly to give an evolutionary angle to the study of cell-substrate adhesions.

Major comments:

1. Figure 1B: Instead of measuring the number of frames, it would be easier to read and reproduce if the results were presented in duration of time (seconds).
2. Since the authors use a Wilcoxon rank-sum test, that compares the medians of the two populations, it would be more sensible to show the medians instead of the means. Same comment for graphs in Figure 3.
3. Supplemental Figure 1: The authors suggest that cell-substrate adhesion duration, regardless of their composition, does not correlate with cell speed. However, these are average durations and average speed. I am not sure the conclusion is solid. One could ask how coherent the durations of the adhesions are within one cell? If durations are very diverse within one cell, then I feel this analysis does not really address the question. If this is the case, the authors may want to put more emphasis on the limitations of their analysis and conclusion.
4. Figure 3: Figure 3A shows that in cells expressing PaxB LIMsOnly, VinculinB does not accumulate at the PaxB LIMsOnly adhesions. I think it would still be interesting to show the dynamics: is there really no accumulation at all, or is there small and transient accumulation? Showing different timepoints with both colours and a quantification of the integrated density of VinB at PaxB LIMsOnly may be informative. In their recent article, the number of PaxB LIMsOnly+/VinB+ adhesions per cell was not 0, yet much lower than in the WT context.
5. In figure 3C-D the difference in duration and lifetime between the PaxB and PaxB LIMsOnly, while statistically significant using the Wilcoxon rank-sum test, appear to me very minimal, with a clear question on whether this very small difference can be biologically relevant.
6. Line 186-187, the authors conclude that the assembly rate is not different (fig 3E) while the p value presented is 0.0003, the same as the p value displayed for fig 3F, for which the authors conclude that they are different. Could they elaborate on this?

minor comment: check a possible typo: line 254: paxB- Dictyostelium cells were (DSC Strain DBS0236728) were obtained.

This is a review by F. Caudron.

Reviewer 2

Comments for the author

In this study, the authors study the dynamics of focal adhesions in *Dictyostelium discoideum*, a model where they have shown before the presence of at least two kinds of adhesions, one where exogenous VinculinB is recruited to adhesions and one where it is not. In the present work, they take a step further to analyze the lifetime & assembly/disassembly rates of the adhesions with and without exogenous VinculinB. The authors describe in detail what is known in the field, which has been mainly studied in mammalian models, highlighting the importance of studying adhesion dynamics and composition in the *Dictyostelium* model. Their findings suggest that the role of VinculinB in *Dictyostelium* is similar to the one in the mammalian adhesions. Moreover, they used a truncated version of PaxillinB that should not bind to VinculinB and found that the adhesions containing the PaxB-LIMs only are more stable than the ones containing PaxB+/VincB+. Thus, they propose a model where other regulatory proteins compensate for the absence of vinculinB in the PaxB-LIMs only adhesions. Their methodology comprises a thorough analysis of the turnover of paxillinB and lifetime of the adhesions using very well established techniques in the field. However, I think two main points should be addressed in order to match the experimental results with the conclusions. One is that the authors state that they are studying the molecular composition of

adhesions, but they are only focusing on VinculinB recruitment to adhesions in the manuscript. The second is that they need to consider the effects of endogenous VinculinB in their phenotype to confidently conclude that the presence/absence of VinculinB controls FAs lifetime/stability. Thus, it would be important to know what happens to the FA dynamics if vinculinB is depleted in this model, since it would be a straightforward strategy to elucidate the real role of vinculinB in Dictyostelium adhesion dynamics.

Specific points:

-In the abstract, line 22, the authors mention that they study the role of molecular composition of the adhesions during cell migration. However, they only focused on the presence/absence of VinculinB, so they should specify this in the sentence to make the abstract clearer.

-The authors mention throughout the manuscript that they analysed paxillinB-containing adhesions that do not contain vinculinB. However, this statement is not entirely accurate since they are not using vinculinB-depleted cells. It is possible that these adhesion structures still contain endogenous vinculinB, which is not detected with the live cell imaging approach. Could the authors test whether the lifetime of adhesions, disassembly rate and paxB turnover change in a vinculinB-null background compared to WT cells? In their previous work, the authors have established that overexpression of paxillinB does not affect adhesion dynamics, so they could overexpress paxillinB and vinculinB in the vinculinB-depleted cells to really pinpoint the role of vinculinB in changing adhesion lifetime and paxillinB turnover at adhesions. These experiments will be valuable to match their conclusions with the experimental data.

-While the authors have described the PaxB-LIMs only construct before, it is not clear from the text or representation in Figure 3B, if the fourth site of paxillinB-vinculinB interaction is in the LIM domains 1 and 2, and therefore, still present in the truncated paxB-LIMs only. Could they please clarify this in the text and figure legend? If indeed, there are still sites of interaction present in paxB-LIMs only, it would be important (if there are reagents available) to show that the truncated N-terminal PaxillinB does not bind to endogenous vinculinB, using an imaging or biochemical approach (for example, staining for endogenous VinculinB in the adhesions containing the truncated PaxillinB or using immunoprecipitation).

-In line 254, delete the word "were" before indicating the Strain of the cells (duplicated at the moment).

Reviewer's Responses to Questions

Experimental quality

Does each figure have the proper controls?

If 'No', please indicate reasons in Comments for Author box below.

Reviewer #1:

- Yes

Reviewer #2:

- Yes

Were the data analyzed using appropriate statistical tests?

If 'No', please indicate reasons in Comments for Author box below.

Reviewer #1:

- Yes

Reviewer #2:

- Yes

Reproducibility

Were experiments performed using adequate number of biological replicates?
If 'No', please indicate reasons in Comments for Author box below.

Reviewer #1:

- Yes

Reviewer #2:

- Yes

Does the methods section provide sufficient detail to permit reproducibility?
If 'No', please indicate reasons in Comments for Author box below.

Reviewer #1:

- Yes

Reviewer #2:

- Yes

Completeness

Are the manuscript's conclusions supported by the data?
If 'No', please indicate reasons in Comments for Author box below.

Reviewer #1:

- Yes

Reviewer #2:

- No

Scholarship

Do the authors cite and discuss the merits of data that would argue for and against their conclusion?
If 'No', please indicate reasons in Comments for Author box below.

Reviewer #1:

- Yes

Reviewer #2:

- Yes

Does the manuscript title & abstract accurately reflect the contents of the manuscript, without hyperbole?

If 'No', please indicate reasons in Comments for Author box below.

Reviewer #1:

- Yes

Reviewer #2:

- No

First revision

Author response to reviewers' comments

We would like to thank the reviewers for their thoughtful reviews and for describing the work as “thorough”, “interesting and well conducted”. Below we respond to each of the reviewer’s comments.

Reviewer 1:

Major comments:

1. *Figure 1B: Instead of measuring the number of frames, it would be easier to read and reproduce if the results were presented in duration of time (seconds).*

→ We have now included time (in seconds) beneath the number of frames on the y-axis.

2. *Since the authors use a Wilcoxon rank-sum test, that compares the medians of the two populations, it would be more sensible to show the medians instead of the means. Same comment for graphs in Figure 3.*

→ Thank you for pointing this out. We now report the medians +/- 95% Confidence Interval (CI) and super-plot the medians for datasets where we use the Wilcoxon rank-sum test to test for statistical significance.

3. *Supplemental Figure 1: The authors suggest that cell-substrate adhesion duration, regardless of their composition, does not correlate with cell speed. However, these are average durations and average speed. I am not sure the conclusion is solid. One could ask how coherent the durations of the adhesions are within one cell? If durations are very diverse within one cell, then I feel this analysis does not really address the question. If this is the case, the authors may want to put more emphasis on the limitations of their analysis and conclusion.*

→ This is an excellent point. We graphed punctae duration to determine the range of punctae duration within an individual cell. These data are shown below, where each column is an individual cell, and we include the individual punctae durations (black outline squares) as well as the mean punctae duration (red circles) for each cell. From this analysis, there is variability in the duration of individual punctae within the same cell, and that the degree of variability in punctae duration within a cell also varies from cell to cell. We include these data as Supplemental Figure 1D and reference the data in the results section. Based on this result, we have adjusted the language in the results text (Lines 150-154) to emphasize the limitations of our conclusions from this analysis and highlight that this does not fully rule out a potential role for punctae lifetime/duration on cell migration speed, albeit likely in concordance with other factors that regulate migration.

4. Figure 3: Figure 3A shows that in cells expressing PaxB LIMsOnly, VinculinB does not accumulate at the PaxB LIMsOnly adhesions. I think it would still be interesting to show the dynamics: is there really no accumulation at all, or is there small and transient accumulation? Showing different timepoints with both colours and a quantification of the integrated density of VinB at PaxB LIMsOnly may be informative. In their recent article, the number of PaxB LIMsOnly+/VinB+ adhesions per cell was not 0, yet much lower than in the WT context.

→ Thank you for raising this point. We intended to live-image VinculinB dynamics and constructed the mApple:VinB strain for this reason. However, while we could visualize punctae, we could not follow the dynamics over time due to photobleaching issues. We adjusted imaging conditions in an effort to circumvent this issue, but could not determine conditions to accurately follow VinB dynamics without photobleaching. As a result, we are unable to provide accurate VinB dynamics. Regarding the quantification of PaxB LIMsOnly+/VinB+ adhesions being a non-zero value in our first publication, this is because we designed our punctae identification pipeline to be semi-automatic with no manual input in the actual detection step. The detection step was based on thresholding and binary selection relative to the mean fluorescence intensity of the entire cell. Given our aforementioned limitations in tracking VinB dynamics, our approach in the previous paper was to evaluate a large number of GFP:PaxillinB-LIMsOnly/mApple:VinculinB expressing *Dictyostelium* cells (n= 552 cells) at single time points to quantify PaxB/VinB status at adhesions. As a result, we would expect to see some noise where our pipeline would call an area of a cell as a “VinB punctae” that overlapped with PaxB-LIMsOnly punctae (which would then be designated as a PaxB LIMsOnly+/VinB+ adhesion), particularly given the large number of cells analyzed. To avoid introducing bias, we did not remove any of these calls from our quantification, hence the non-zero (but very low) value.

5. In figure 3C-D the difference in duration and lifetime between the PaxB and PaxB LIMsOnly, while statistically significant using the Wilcoxon rank-sum test, appear to me very minimal, with a clear question on whether this very small difference can be biologically relevant.

→ We agree that this question is important to speculate on in the manuscript. As pointed out by the reviewer, the difference in duration and lifetime between PaxB and PaxB LIMsOnly, while statistically significant, is modest, with an increased median of ~10% in both duration and lifetime for PaxB LIMsOnly relative to wildtype PaxB. This difference stands in contrast to some of the larger differences observed in literature investigating mammalian adhesion lifetime upon Paxillin truncation (~50% decrease in lifetime of PaxB LIMs relative to PaxB WT; PMID: 33782527) or perturbation/knockout of other adhesion components such as Talin perturbation (~55% decrease vs WT talin; PMID: 33783351) or FAK perturbation (~200% increase vs WT FAK; PMID: 22270917). However, it has been additionally shown that adhesion composition or perturbation of adhesion molecules do not solely define adhesion dynamics, with other factors playing a role, such as adhesion orientation (PMID: 31320676), adhesion location (PMID: 31320676), mechanical forces (PMID: 20613844), actomyosin contractility and actin dynamics (PMIDs: 18800171, 28468976), adhesion maturation stage (PMID: 21689521) and extracellular factors such as substrate stiffness (PMID: 28468976). Thus, we believe that PaxB truncation is one of many factors that regulate adhesion dynamics during cell migration.

Additionally, given the fast speeds of migrating *Dictyostelium* cells and the significantly shorter lifetime of *Dictyostelium* adhesions (~15-20 sec; PMID: 40498666) compared to mammalian and other vertebrate models (usually in the range of minutes; PMIDs: 36723624, 33782527, 26842895, 26903539, 21779367, 22291038), we postulate that even modest changes in adhesion lifetime that we observe could have biologically-relevant consequences for *Dictyostelium*, but acknowledge more work is needed to test this hypothesis. We incorporate both of these thoughts in our discussion (Lines 281-300).

6. Line 186-187, the authors conclude that the assembly rate is not different (fig 3E) while the *p* value presented is 0.0003, the same as the *p* value displayed for fig 3F, for which the authors conclude that they are different. Could they elaborate on this?

→ The assembly rates are statistically different - We have corrected this error in the text.

7. minor comment: check a possible typo: line 254: *paxB*- *Dictyostelium* cells were (DSC Strain DBS0236728) were obtained.

→ Thank you for catching this typo.

Reviewer 2:

Specific points:

1. In the abstract, line 22, the authors mention that they study the role of molecular composition of the adhesions during cell migration. However, they only focused on the presence/absence of VinculinB, so they should specify this in the sentence to make the abstract clearer.

→ We have adjusted the language to make it clear that we are focusing on the presence/absence of VinculinB.

2. The authors mention throughout the manuscript that they analysed *paxillinB*-containing adhesions that do not contain vinculinB. However, this statement is not entirely accurate since they are not using vinculinB-depleted cells. It is possible that these adhesion structures still contain endogenous vinculinB, which is not detected with the live cell imaging approach. Could the authors test whether the lifetime of adhesions, disassembly rate and *paxB* turnover change in a vinculinB-null background compared to WT cells? In their previous work, the authors have established that overexpression of *paxillinB* does not affect adhesion dynamics, so they could overexpress *paxillinB* and vinculinB in the vinculinB-depleted cells to really pinpoint the role of vinculinB in changing adhesion lifetime and *paxillinB* turnover at adhesions. These experiments will be valuable to match their conclusions with the experimental data.

→ This is an important point, and one that we are unable to experimentally address. There are only a small number of other papers on *Dictyostelium* VinculinB (PMIDs: 22391412, 22588127).

Therefore, at this time there are limited available resources: There is no VinculinB mutant available in the *Dictyostelium* resource databases (Dictybase and the NBRP Nenkin) nor are there VinculinB antibodies. Thus, we are unable to compare PaxillinB dynamics in a VinculinB-null background. With this in mind, we have adjusted our language in the discussion text to raise the caveat regarding the presence of endogenous VinculinB. These text adjustments are in the discussion section in lines 232-234.

3. While the authors have described the *PaxB-LIMs* only construct before, it is not clear from the text or representation in Figure 3B, if the fourth site of *paxillinB*-vinculinB interaction is in the LIM domains 1 and 2, and therefore, still present in the truncated *paxB-LIMs* only. Could they please clarify this in the text and figure legend? If indeed, there are still sites of interaction present in *paxB-LIMs* only, it would be important (if there are reagents available) to show that the truncated N-terminal PaxillinB does not bind to endogenous vinculinB, using an imaging or biochemical approach (for example, staining for endogenous VinculinB in the adhesions containing the truncated PaxillinB or using immunoprecipitation).

→ This reviewer is correct. The predicted “4th interaction site” is maintained in the *PaxB-LIMs*Only truncation, and we have made this point clearer in the text and figure legend. However, based on the fact that we do observe co-localization of VinculinB with PaxillinB when the PaxillinB N-terminus is truncated (the *PaxillinB-LIMs*Only molecule) we hypothesize that the predicted interactions sites 1-3 on PaxillinB are required for initial interactions with VinculinB, with interactions with the PaxillinB LIM domains then serving to further stabilize the binding axis at adhesions after the initial binding interaction is established. Additionally, previous work in

mammalian cell lines shows that mammalian Vinculin binds to the N-terminal portions of Paxillin, specifically in the conserved LD motifs (PMIDs: 7525621, 8922390). Thus, we predict that ablating sites 1-3 in the PaxillinB N-terminus perturb VinculinB interactions, however, we cannot exclude the possibility of minimal interactions beyond the limit of detection by light microscopy. As we indicate in point 2 above, there are no available VinculinB antibodies to test PaxB/VinB interactions biochemically. Instead, we have adjusted language in the text and figure legend to incorporate the interpretations we included above.

4. In line 254, delete the word "were" before indicating the Strain of the cells (duplicated at the moment).

→ Thank you for catching this typo - it has been corrected.

Second decision letter

MS ID#: bio.062197R1

MS Title: Differential PaxillinB dynamics at Dictyostelium cell-substrate adhesions

Authors: JULIO Cesar FIERRO MORALES; Minna Roh-Johnson

Dear Dr Roh-Johnson,

I am happy to tell you that your manuscript has been accepted for publication in Biology Open, pending our standard publication integrity checks. It was accepted on 05 Sep 2025.

To see the reviewers' reports and a copy of this decision letter, please go to: View Reviewer Comments